# Out-of-pocket expenditure on medicines in Bangladesh: An analysis of the national household income and expenditure survey 2016–17

Edson Serván-Mori[1], Md Deen Islam[2], Warren A. Kaplan[3], Rachel Thrasher[2], Veronika J. Wirtz[3]*

1 Center for Health Systems Research, National Institute of Public Health, Cuernavaca, Morelos, Mexico,
2 Global Development Policy Center, Boston University, Boston, Massachusetts, United States of America,
3 Department of Global Health, Boston University, School of Public Health, Boston, Massachusetts, United States of America

* vwirtz@bu.edu

**Data Availability Statement:** https://catalog.ihsn.org/index.php/catalog/7399.

## Abstract

### Background and objectives

High out-of-pocket expenditures (OOPE) increases the probability that households will become impoverished or will forgo needed care. The aim of this paper is to study household medicines expenditure and its associated determining factors to develop policies to protect households from financial hardship.

### Methods

The present cross-sectional and population-level study used the Bangladesh 2016–17 National Household Income and Expenditure Survey (HIES). The final sample size was 46,080 households. We analyzed the probability of OOPE for medicines, the share of total OOPE due to medicines out of total OOPE in health (reported as a ratio between zero and one), the OOPE amount for medicines reported (in United States Dollars), and the share of OOPE amount on medicines out of total household expenditure (reported as a ratio between zero and one). Predictors of analyzed outcomes were identified using three regression models.

### Results

Out of those households who spent on healthcare, the probability of having any OOPE on medicines was 87.9%. Of those who spent on medicines, the median monthly expenditure was US$3.03. The poorest households spent 9.97% of their total household expenditure as OOPE on medicines, nearly double that of the wealthiest households (5.86%). The characteristic which showed the most significant correlation to a high OOPE on medicines was the presence of chronic diseases, especially cancer. Twenty six percent of all surveyed households spend more than 10% of their OOPE on medicines.

**Funding:** The authors declare support from the Rockefeller Brothers Foundation, Grant # 21-176.

**Competing interests:** The authors have declared that no competing interests exist.

## Conclusions

Our study shows that financial protection should be targeted at the poorest quintiles and such protection should include enrollment of rural households. Further, outpatient medicines benefits should include those for non-communicable diseases (NCDs).

## Introduction

One of the most important concerns in low- and middle-income countries (LMICs) is the increasing out-of-pocket health care expenditure (OOPE) made by households and individuals. OOPE is the amount of money paid by households to purchase health services and medicines when members of a household have a health care need. Healthcare costs are among the largest barriers to accessing health services and achieving universal health coverage (UHC), and among the most important factors associated with the reduction of the welfare of households. In particular, high OOPE increases the probability that households will become impoverished or will forgo needed care [1] and, as a result, households may decide to sell assets or take out loans to pay for this healthcare [2, 3].

Out of all possible healthcare financing mechanisms, OOPE is considered the most inequitable. It has been estimated that, due to health expenditure, 100 million households fall into extreme poverty every year -living on US$1.90 per day or less [4]. In this regard, in many countries medicines represent the largest proportion of OOPE on healthcare. A financial protection analysis in eight Southeast Asian countries showed that in seven out of eight countries medicines represent between 75–81% of OOPE [5]. Despite significant progress towards achieving UHC in many low- and middle-income countries, substantial challenges remain in terms of access to quality healthcare and lack of financial protection [6, 7].

Out of pocket expenditure is amendable by public policy. For instance, the implementation of health insurance should protect households from large OOPE, including medicines. Furthermore, policies regulating the payment of providers have an influence on their behavior, including ordering diagnostic tests, prescribing medicines and recommending surgery and other types of treatment. These considerations make the study of OOPE relevant for setting health policies and assessing their effect on economic development and poverty reduction. There is ample literature on the study of OOPE [8], the methodological foundation [9] and its current application to assess progress on UHC [1].

Evidence shows that in many countries the majority of health OOPE is for medicines. For instance, in India 90% of OOPE on health is on medicines, in Nepal it is 88%, and Indonesia it is 78%. Medicines OOPE are also important as a proportion of total household expenditure. According to the World Health Survey, up to 95% of the total expenditure of poorer households in LMICs is spent on medicines, and this is far higher than the 3.5% expended by poorer households in high income countries (HICs) [10]. Approximately half (41%-56%) of households in LMICs spent 100% of their health care expenses on medicines [10]. Medicines OOPE is a large driver of overall OOPE and countries have made a commitment to UHC. Thus, studies focusing on medicines OOPE have become increasingly relevant.

Bangladesh is a country with over 160 million inhabitants and rapid urbanization that has made great progress in health, education, and economic development over the past decades. Maternal mortality has fallen by 60% over the past two decades and child mortality by two-thirds [11]. Bangladesh is expected to 'graduate' from the World Trade Organization's designation as a 'least developed country' as soon as 2024 [12]. However, Bangladesh faces

significant challenges to improve the health and wellbeing of its population due to a lack of coherent social security or financial health protection. Bangladesh is incurring a demographic shift toward longer life expectancy and it is experiencing an epidemiological transition from predominantly infectious diseases toward chronic, non-communicable diseases that require sustained medication and life-long treatment. For example, the prevalence of diabetes in Bangladesh is relatively high at about 10% [13]. Most patients with diabetes require long-term medication, diagnostic and monitoring devices apart from other medical care. We have previously shown, *ex ante*, that household OOPE on insulin is likely to have an effect on the probability of individual households' becoming impoverished, as well as having a much more expansive effect on the country's welfare [14].

Bangladesh, however, spends only a relatively low percentage of GDP on health compared to several other countries in the region and relies heavily on OOPE as the main source of health financing. Just 2.27% of Bangladesh's GDP is spent on health [15] out of which 74% is OOPE [5] and medicines are the largest component of OOPE in health. The 2005 National Household and Expenditure Survey identified the cost of medicine as greater than any other factor in determining OOPE on health [16]. The 2010 Bangladesh National Household and Expenditure Survey showed that medicines represented 61% of the OOPE on health [17] and a recent cross-sectional survey showed that prices of some essential medicines in Bangladesh are consistently expensive across both public and private sector facilities [18].

The contributions of this present study are twofold. First, there is a gap in our understanding of medicines OOPE determinants. Knowing these determinants would support the development of policies to protect households from financial hardship. Second, this study uses the most recent national-level survey data on healthcare utilization and OOPE for out-patient in Bangladesh, making the study highly relevant from public policy standpoint. The aim of this paper is to study household medicines expenditure and its associated determining factors.

## Material and methods

### Settings/design and analytical sample

The present cross-sectional and population-level study used the Bangladesh 2016–17 National Household Income and Expenditure Survey (HIES). Details of the survey design have been described elsewhere [19]. Briefly, HIES seeks to obtain detailed data on household income, expenditure and consumption, determine the poverty profile with urban and rural breakdown and district-level poverty, provide household level consumption data for compiling national accounts estimates, and provide relevant data for monitoring of the Poverty Reduction Strategy, five-year plan and the Sustainable Development Goals. This survey contains a wide range of socio-economic information at the household level that has strong influence on the decision-making process for the government.

The sample for the HIES is explicitly designed to produce estimates at the three levels (urban and rural, district-level, and household level) and is designed with an urban sample size large enough to understand Bangladesh's urbanization patterns. A sample design was adopted for the HIES with 2,304 Primary Sampling Units (PSU) in eight administrative and geographical divisions (Barisal, Chittagong, Dhaka, Khulna, Mymensingh, Rajshahi, Rangpur and Sylhet) and 64 districts, selected from the last Housing and Population Census (2011) [20]. Within each PSU, 20 households were selected for interviews. The final sample size was 46,080 households [20] and was stratified at the district level, including a total of 132 sub-strata: 64 urban, 64 rural, and four main City Corporations.

For the present study, we excluded 3.1% of households surveyed with incomplete relevant information, the analytical sample included 43,659 households (representing all approximately

37.6 million households in Bangladesh). To test whether there is a difference between the included and the excluded households, we examined potential differences in covariates that could be associated with our outcome variables between our analytical sample and those excluded ones. We did not find any significant differences.

## Variables

From our sample (see above), we analyzed five main outcome variables: 1) The probability of OOPE in medicines, 2) the share of medicines OOPE out of total OOPE in health (reported as a ratio between zero and one), 3) the OOPE amount in medicines reported (in United States dollars based on the annual average exchange rate obtained from the central bank of Bangladesh), 4), the share of OOPE on medicines out of total household expenditure (reported as a ratio between zero and one)- based on the method of Wagstaff et al. [2, 8], and 5) the share of OOPE on medicines out of a given household's capacity to pay (reported also as a ratio between zero and one)- based on the method of Xu et al. [9, 21]. We calculated the amount of total OOPE in health by taking the sum of all expenditures reported by households in the last month and year before the survey, including out- and inpatient care and medicines.

We followed previous studies [17, 21, 22] and included as covariates *head of household characteristics* such as age, sex (male = 1/female = 0), schooling (none, primary, secondary and tertiary), religion (Islam, Hinduism, Buddhism, other), marital status (married, never married, widowed/Divorced/Separated), and working during the last seven days. We also included h*ousehold characteristics* as covariates: number of 'equivalent adults'which adjusts for the economy of scale in consumption. That is, a household with three members, including children, does not consume three times that of a one-person household. According to the OECD [23], the equivalence scale considers the age of the household members and establishes a standardization that allows comparison). We also considered as covariates the demographic dependence ratio (the number of dependents aged 0–14 and those over the age of 65, compared with the total population aged 15–64), an additive and unweighted disability index (measured by the presence in all household members of difficulty in seeing, hearing, walking, climbing, remembering or concentrating, washing all over or dressing and communicating) operationalized as a percentage, the presence (yes = 1/no = 0) of any member with any symptoms of illness/injury in the last 30 days, the use of health services (yes = 1/no = 0) by any of the members who reported a health need in the last 30 days, the presence (yes = 1/no = 0) of any member with a chronic disease (categorized as diabetes, cardiovascular disease, cancer, chronic diseases of infectious origin, disabilities, others), an asset and housing material-index as a measure of socioeconomic status constructed using factor analysis [24, 25] and expressed as a percentage, the participation in any safety nets or social programs (yes = 1/no = 0) and place of residence such as rural/urban and the administrative and geographical division.

## Statistical analysis

We used survey weights to account for the complex survey design in all descriptive and multivariable analyses. We report population estimates for all results. All analyses were performed using the *svy* module of the statistical package Stata version 15.1 [26]. We first quantified the household characteristics described previously reporting mean, percentage and their 95% confidence intervals (CIs). The median of the expenditure was calculated for those households which had an expenditure greater than zero.

We developed four main outcome variables among surveyed households as a function of the quintile of monthly household expenditure per equivalent adult. We report median and interquartile range, and percentage and their 95% CI.

Predictors of analyzed outcome variables were estimating with three regression models: 1) a logistic regression model [27] for the probability of OOPE on medicines; 2) three fractional logistics multiple regression models [28, 29] for the share of OOPE on medicines out of: OOPE on health; total household expenditure; a given household's capacity to pay (the latter was estimated using the STATA *fracreg*); and 3) a linear regression model [27, 28] for the OOPE on medicines per adult equivalent expressed as a napierian logarithm. For the first regression model, we reported adjusted odds-ratios (aORs) and CI95% and for the second and third models, we reported adjusted coefficients (aCoeff) and CI95%. Finally, based on the first regression analysis results, we estimated the adjusted share of OOPE on medicines out of total household expenditure by considering the following thresholds of the total household expenditure: 10, 15, 20, 30%). We also used the first regression model to estimate share of medicine OOPE out of a household's capacity to pay by considering different thresholds (10, 15, 20, 30 and 40%) of the total household expenditure) [30] as well as according to the quintile of household expenditure per equivalent adult.

Following previous studies [22], our estimations had to consider the presence of a selection bias related to the decision to spend funds on health because there may be particular household characteristics that increase the probability of health expenditure. This bias applied to all households. Following Heckman (1979) [31], we used a logistic model to estimate the conditional probability that a given household would record any given health OOPE (as a function of the household characteristics mentioned above). We then calculated the Mills ratio [32] to capture the magnitude of the selection bias for each household analyzed. Subsequently, this parameter was incorporated as a regressor in the regression models described above.

## Results

Table 1 describes the household characteristics. The mean age of household heads was 43.6 years and 87.7% of household heads were male. About 75% of all household heads have either no or only a primary level schooling. Nine out of ten household heads report Islam as their religion and 91.8% of the household heads reported being married. Nearly half (46.1%) of households had at least one member with at least one chronic disease (e.g. cardiovascular diseases, diabetes, cancer). Half of the households reported having a member with symptoms of illness/injury in the last 30 days and out of this half, 89.4% used health services. One in five households (21.2%) are benefiting from a social program.

Most surveyed households are in rural areas (71.3%), with only 25.3% in the urbanized Dhaka area. There are large disparities between the lowest quintile of monthly household expenditure (Quintile 1) and the wealthiest (Quintile 5), in particular regarding the demographic dependence (95.5 and 62.8 respectively) and socioeconomic index (11.9 and 32.7 respectively).

Among all households, the probability of any OOPE on health in the last month was 74.4%. Out of those households who spent anything on healthcare, the probability of having OOPE on medicines was 87.9% (Table 2). When we adjusted per equivalent adult, out of those households who spent anything on healthcare, their median total monthly health-related OOPE was US$3.1. With similar adjustment per equivalent adult, those households spending OOP on medicines had a median OOPE of US$3.0. Thus, 96.5% of the total monthly healthcare-related OOPE was on medicines.

The OOPE share out of total household expenditure on healthcare and medicines was 8.2% and 8.1%, respectively. In this regard, households in the first income quintile (poorest households) spent 8.9% of their OOPE on healthcare as a share of their total household expenditure whereas those in the 5th income quintile (wealthiest) spent only 6.7%. With regard to

**Table 1. Main household characteristics.** HIES, Bangladesh, 2016/17.

| Sample size (n) = 43,659 households | Mean or % and CI 95% |
|---|---|
| weighted sample (N) = 37,616,656 households | |
| *Household head* | |
| Age (yrs) | 43.57 [43.27–43.86] |
| Male | 87.67 [87.10–88.24] |
| Schooling | |
| Nothing | 30.59 [29.61–31.56] |
| Primary | 43.28 [42.27–44.30] |
| Secondary | 21.22 [20.27–22.16] |
| Tertiary | 4.91 [4.35–5.48] |
| Religion | |
| Islam | 89.29 [87.72–90.87] |
| Hinduism | 9.48 [8.22–10.74] |
| Buddhism | 0.88 [0.32–1.44] |
| Other | 0.35 [0.19–0.50] |
| Marital status | |
| Marriage | 91.78 [91.42–92.14] |
| Never marriage | 2.10 [1.91–2.28] |
| Widowed/Divorced/Separated | 6.12 [5.81–6.44] |
| Working | 84.15 [83.34–84.96] |
| *Household* | |
| Equivalent adults | 2.73 [2.71–2.75] |
| Demographic dependence | 79.30 [77.95–80.65] |
| Disability index | 63.21 [58.31–68.12] |
| Any member with any symptoms of illness/injury in the last 30 days | 53.32 [51.93–54.72] |
| Use of health services | 89.32 [87.72–90.91] |
| Any member with a chronic disease | 46.15 [44.77–47.53] |
| Often infectious origin | 60.45 [59.18–61.72] |
| Disabilities | 34.55 [33.29–35.81] |
| Diabetes | 12.12 [11.22–13.02] |
| Cardiovascular disease | 28.76 [27.61–29.91] |
| Cancer | 0.53 [0.41–0.65] |
| Others chronic disease | 18.85 [17.92–19.78] |
| Socioeconomic index | 19.39 [18.77–20.01] |
| Beneficiary/member of any safety nets/social program | 21.18 [20.07–22.29] |
| *Place of residence* | |
| Rural | 71.30 [69.03–73.57] |
| Division | |
| Barisal | 5.57 [5.19–5.95] |
| Chittagong | 19.98 [17.86–22.11] |
| Dhaka | 25.30 [22.77–27.83] |
| Khulna | 10.90 [10.14–11.67] |
| Mymensingh | 7.59 [6.18–9.00] |
| Rajshahi | 12.55 [11.04–14.06] |
| Rangpur | 10.95 [9.83–12.07] |
| Sylhet | 7.15 [6.69–7.62] |

**Table 2. OOPE on health and medicines according to quintile of household expenditure.** HIES, Bangladesh, 2016.

| Weighted sample (N) = 37,616,656 households | Overall | Quintile of monthly household expenditure per equivalent adult | | | | |
|---|---|---|---|---|---|---|
| | | 1st | 2nd | 3th | 4th | 5th |
| *Total household expenditure per equivalent adult (US $), p50 and IQR* | 60.42 [41.59–92.24] | 30.87 [25.92–34.73] | 44.97 [41.59–48.47] | 60.42 [56.31–65.20] | 83.20 [76.18–92.24] | 145.36 [120.99–192.29] |
| Probability of OOPE on health, % | 74.41 [72.93–75.89] | 70.24 [68.40–72.08] | 75.85 [73.88–77.81] | 74.13 [71.84–76.41] | 73.42 [71.09–75.76] | 78.42 [75.73–81.11] |
| *OOPE on health per equivalent adult (US$), p50 and IQR* | 3.14 [1.12–8.77] | 1.79 [0.66–4.52] | 2.57 [0.94–6.18] | 3.21 [1.22–8.26] | 4.25 [1.51–11.81] | 5.47 [1.74–16.49] |
| Share of OOPE on health out-off household expenditure, % | 8.20 [7.90–8.50] | 8.87 [8.39–9.34] | 8.63 [8.19–9.08] | 8.29 [7.80–8.78] | 8.36 [7.76–8.96] | 6.86 [6.32–7.41] |
| Share of inpatient expenditure out-off OOPE on health, % | 4.36 [3.99–4.73] | 2.78 [2.33–3.24] | 3.70 [3.13–4.27] | 4.26 [3.26–5.26] | 5.15 [4.26–6.05] | 5.76 [4.89–6.63] |
| Share of outpatient expenditure out-off OOPE on health, % | 95.64 [95.27–96.01] | 97.22 [96.76–97.67] | 96.30 [95.73–96.87] | 95.74 [94.74–96.74] | 94.85 [93.95–95.74] | 94.24 [93.37–95.11] |
| Probability of OOPE on medicines, % | 87.92 [86.63–89.21] | 86.96 [85.75–88.17] | 88.36 [87.05–89.68] | 88.49 [87.07–89.90] | 89.24 [87.77–90.70] | 86.57 [82.32–90.81] |
| *OOPE on medicines per equivalent adult (US$), p50 and IQR* | 3.03 [1.32–7.32] | 1.80 [0.85–4.09] | 2.42 [1.12–5.31] | 3.06 [1.41–6.84] | 3.82 [1.65–9.31] | 5.25 [2.12–13.06] |
| Share of OOPE on medicines out-off OOPE on health, % | 71.49 [70.24–72.73] | 74.96 [73.70–76.22] | 73.85 [72.44–75.27] | 71.83 [70.26–73.39] | 70.52 [68.89–72.14] | 66.67 [63.00–70.33] |
| Share of OOPE on medicines out-off household expenditure, % | 8.06 [7.80–8.33] | 9.97 [9.46–10.47] | 8.59 [8.18–9.00] | 8.13 [7.69–8.57] | 7.98 [7.49–8.48] | 5.86 [5.32–6.39] |
| Share of OOPE on medicines out-off household's capacity to pay, % | 14.16 [13.96–14.36] | 19.85 [19.30–20.40] | 16.87 [16.42–17.33] | 14.42 [13.99–14.85] | 11.74 [11.36–12.11] | 7.60 [7.32–7.89] |

medicines, the proportion of total household expenditure relegated to medicine OOPE is the highest for the poorest households (9.9%)–nearly double that of the wealthiest households (5.9%). Expressed as the capacity of the household to pay, the disparity between households in the poorest and the wealthiest quintile is even larger: while poor households spent nearly a one-fifth (19.8%) of their disposable income on medicines, the wealthiest households spent 7.6% on medicines.

Moreover, the overwhelming proportion of the healthcare OOPE is for outpatient expenditures (95.6% outpatient versus 4.3% inpatient). See Table 2.

Several factors are associated with the increased probability of having any OOPE for medicines in a household, including for instance age of the household head, the household head never being married, being in a rural area, and reporting a chronic disease (except cancer) compared to those not having the disease (Table 3). By contrast, having a male household head, having a job, and being a beneficiary of a social program reduced the probability of OOPE on medicines. The actual amount of medicines expenditure (adjusted per number of equivalent adults in the household) increased with factors including the age of the household heads, the household head education, households with larger demographic dependence, living in rural areas, and with all chronic diseases (especially cancer). Being covered by a safety net/social program reduces the share of OOPE on medicines out of total health expenditure, reduces the amount of OOPE on medicines and the share of OOPE on medicines out of household expenditure.

The characteristic which showed the most significant association with a high OOPE on medicines relative to total household expenditure was the use of health services and the presence of chronic diseases, especially cancer, followed by diabetes and cardiovascular diseases. The amount, as well as the share, of OOPE on cancer medicines out of the total household expenditure has by far the highest incremental risks (Fig 1).

**Table 3. Factors associated with OOPE on medicines.** HIES, Bangladesh, 2016/17.

| | Logistic regression model: OOPE on medicines > 0 | Factional regression model: Share of OOPE on medicines out-off OOPE on health | OLS regression model: OOPE on medicines per adult equivalent (ln) | Factional regression model: Share of OOPE on medicines out-off total household expenditure or household's capacity to pay | |
|---|---|---|---|---|---|
| | | | | out-off total household expenditure | out-off household's capacity to pay |
| | *Adjusted odds-ratios* | *Adjusted coefficient* | *Adjusted coefficient* | *Adjusted coefficient* | |
| *Household head* | | | | | |
| Age (yrs) | 1.036 [1.027−1.045]*** | 0.017 [0.014−0.020]*** | 0.010 [0.008−0.012]*** | 0.005 [0.003−0.007]*** | 0.004 [0.002−0.006]*** |
| Male | 0.543 [0.362−0.816]** | -0.380 [-0.508−-0.252]*** | 0.107 [0.026−0.187]** | 0.114 [0.018−0.210]* | 0.146 [0.046−0.245]** |
| Schooling | | | | | |
| Nothing | Ref. | Ref. | Ref. | Ref. | Ref. |
| Primary | 0.779 [0.633−0.958]* | -0.173 [-0.248−-0.097]*** | 0.116 [0.069−0.164]*** | -0.058 [-0.111−-0.004]* | -0.093 [-0.147−-0.038]** |
| Secondary | 0.668 [0.516−0.866]** | -0.338 [-0.434−-0.242]*** | 0.231 [0.161−0.301]*** | -0.071 [-0.152−0.009]+ | -0.167 [-0.248−-0.085]*** |
| Tertiary | 0.697 [0.472−1.029]+ | -0.439 [-0.588−-0.291]*** | 0.304 [0.208−0.400]*** | -0.126 [-0.256−0.004]+ | -0.264 [-0.398−-0.129]*** |
| Religion | | | | | |
| Islam | Ref. | Ref. | Ref. | Ref. | Ref. |
| Hinduism | 0.857 [0.624−1.178] | -0.043 [-0.136−0.049] | -0.098 [-0.167−-0.030]** | -0.061 [-0.142−0.019] | -0.013 [-0.084−0.058] |
| Buddhism | 0.601 [0.342−1.054]+ | 0.108 [-0.408−0.623] | -0.287 [-0.646−0.071] | -0.292 [-0.646−0.063] | -0.203 [-0.568−0.162] |
| Other | 0.623 [0.193−2.007] | -0.087 [-0.423−0.248] | -0.078 [-0.309−0.153] | -0.196 [-0.442−0.050] | -0.168 [-0.423−0.087] |
| Marital status | | | | | |
| Marriage | Ref. | Ref. | Ref. | Ref. | Ref. |
| Never marriage | 3.842 [2.109−6.997]*** | 0.731 [0.510−0.952]*** | 0.106 [-0.041−0.253] | 0.196 [0.041−0.352]* | 0.099 [-0.045−0.244] |
| Widowed/Divorced/Separated | 1.027 [0.608−1.734] | 0.008 [-0.146−0.161] | -0.252 [-0.344−-0.160]*** | 0.026 [-0.069−0.121] | 0.064 [-0.036−0.163] |
| Working | 0.741 [0.525−1.046]+ | 0.081 [-0.016−0.178] | -0.179 [-0.246−-0.112]*** | -0.220 [-0.294−-0.146]*** | -0.124 [-0.202−-0.047]** |
| *Household* | | | | | |
| Equivalent adults | 0.956 [0.822−1.111] | -0.106 [-0.142−-0.069]*** | | -0.129 [-0.158−-0.101]*** | -0.132 [-0.160−-0.105]*** |
| Demographic dependence | 0.878 [0.762−1.013]+ | 0.035 [-0.012−0.082] | 0.015 [-0.019−0.048] | 0.047 [0.010−0.083]* | 0.110 [0.075−0.146]*** |
| Disability index | 0.988 [0.932−1.048] | 0.005 [-0.012−0.022] | 0.020 [0.009−0.032]** | 0.030 [0.020−0.041]*** | 0.031 [0.019−0.043]*** |
| Any member with any symptoms of illness/injury in the last 30 days | 0.978 [0.630−1.519] | -0.198 [-0.390−-0.007]* | -0.148 [-0.271−-0.026]** | -0.187 [-0.318−-0.055]** | -0.188 [-0.322−-0.055]** |
| Use of health services | 10.322 [5.828−18.280]*** | -0.521 [-0.711−-0.331]*** | 0.668 [0.539−0.797]*** | 0.419 [0.294−0.544]*** | 0.492 [0.365−0.620]*** |
| Chronic disease | | | | | |
| Often infectious origin | 3.798 [2.705−5.334]*** | 0.173 [0.118−0.227]*** | 0.295 [0.252−0.338]*** | 0.185 [0.142−0.229]*** | 0.160 [0.115−0.205]*** |
| Disabilities | 2.595 [1.801−3.739]*** | 0.146 [0.081−0.211]*** | 0.344 [0.297−0.391]*** | 0.262 [0.213−0.310]*** | 0.265 [0.214−0.316]*** |
| Diabetes | 3.780 [1.482−9.639]** | 0.168 [0.038−0.297]* | 0.652 [0.586−0.717]*** | 0.333 [0.261−0.406]*** | 0.300 [0.225−0.376]*** |

*(Continued)*

**Table 3.** (Continued)

| | Logistic regression model: OOPE on medicines > 0 | Factional regression model: Share of OOPE on medicines out-off OOPE on health | OLS regression model: OOPE on medicines per adult equivalent (ln) | Factional regression model: Share of OOPE on medicines out-off total household expenditure or household's capacity to pay | |
|---|---|---|---|---|---|
| | | | | out-off total household expenditure | out-off household's capacity to pay |
| | *Adjusted odds-ratios* | *Adjusted coefficient* | *Adjusted coefficient* | *Adjusted coefficient* | |
| Cardiovascular disease | 6.022 [3.896−9.310]*** | 0.263 [0.192−0.334]*** | 0.503 [0.448−0.557]*** | 0.319 [0.264−0.373]*** | 0.322 [0.263−0.381]*** |
| Cancer | 4.481 [0.747−26.875] | -0.248 [-0.594−0.097] | 1.497 [1.145−1.849]*** | 1.205 [0.813−1.598]*** | 1.031 [0.603−1.460]*** |
| Others chronic disease | 3.811 [2.643−5.494]*** | 0.070 [-0.007−0.148]+ | 0.663 [0.606−0.720]*** | 0.520 [0.462−0.579]*** | 0.431 [0.373−0.490]*** |
| Socioeconomic index | 0.984 [0.978−0.991]*** | -0.010 [-0.012−-0.007]*** | 0.006 [0.004−0.008]*** | -0.010 [-0.012−-0.008]*** | -0.016 [-0.018−-0.014]*** |
| Beneficiary/member of any safety nets/social program | 0.732 [0.626−0.855]*** | -0.107 [-0.160−-0.054]*** | -0.018 [-0.061−0.025] | -0.038 [-0.084−0.008] | -0.041 [-0.085−0.004]+ |
| *Place of residence* | | | | | |
| Rural | 0.990 [0.738−1.329] | 0.089 [0.020−0.158]* | 0.077 [-0.005−0.160]+ | 0.107 [0.047−0.167]*** | 0.146 [0.087−0.205]*** |
| *Division* | | | | | |
| Barisal | Ref. | Ref. | Ref. | Ref. | Ref. |
| Chittagong | 1.688 [1.182−2.411]** | -0.073 [-0.158−0.013]+ | 0.214 [0.122−0.305]*** | -0.015 [-0.095−0.065] | 0.041 [-0.038−0.120] |
| Dhaka | 0.860 [0.553−1.338] | 0.025 [-0.072−0.121] | -0.170 [-0.279−-0.062]** | -0.328 [-0.411−-0.245]*** | -0.284 [-0.364−-0.203]*** |
| Khulna | 0.382 [0.289−0.507]*** | -0.372 [-0.450−-0.295]*** | -0.514 [-0.610−-0.418]*** | -0.346 [-0.420−-0.272]*** | -0.200 [-0.269−-0.131]*** |
| Mymensingh | 0.878 [0.532−1.450] | 0.222 [0.084−0.360]** | -0.427 [-0.549−-0.306]*** | -0.366 [-0.484−-0.247]*** | -0.090 [-0.201−0.021] |
| Rajshahi | 0.298 [0.219−0.405]*** | -0.373 [-0.457−-0.288]*** | -0.518 [-0.614−-0.422]*** | -0.295 [-0.371−-0.219]*** | -0.125 [-0.198−-0.052]** |
| Rangpur | 0.959 [0.684−1.344] | 0.195 [0.104−0.287]*** | -0.436 [-0.537−-0.335]*** | 0.002 [-0.078−0.082] | 0.124 [0.051−0.198]** |
| Sylhet | 1.783 [1.113−2.857]* | 0.307 [0.197−0.417]*** | -0.149 [-0.251−-0.047]** | -0.263 [-0.349−-0.177]*** | -0.069 [-0.150−0.012]+ |
| Intercept | 6.882 [3.395−13.952]*** | 1.966 [1.753−2.180]*** | 0.845 [0.677−1.014]*** | -2.272 [-2.441−-2.104]*** | -1.759 [-1.928−-1.591]*** |
| Weighted sample (N) | 27,991,346 | 27,991,346 | 24,609,301 | 27,991,346 | 27,322,598 |

***P<0.001,

**P<0.01,

*P<0.05,

+P<0.10.

In order to capture the magnitude of the selection bias in the occurrence of household expenditure in health, five models were adjusted also by Mills ratio.

Twenty six percent of households spend more than 10% of their OOPE on medicines. By increasing the spending threshold to 15% of OOPE, the percentage of households incurring this level of spending dropped to 6.8% (Table 4). Nearly two percent (1.8%) of households spent 20%, and 0.1% spent 30% of their total expenditure on OOPE for medicines. These proportions decrease with increasing wealth. For instance, 27.3% of households in the 1st quintile spent 10% of their household expenditure on medicines compared to 23.6% in the 5th quintile.

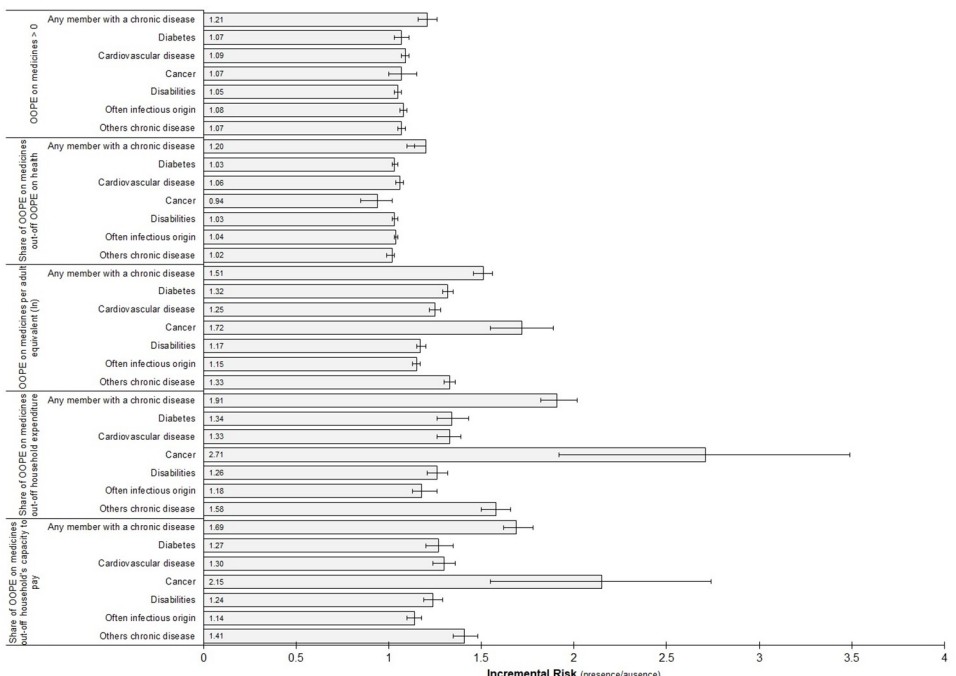

**Fig 1. Adjusted incremental risk of OOPE on medicines according to household presence chronic disease.** HIES, Bangladesh, 2016.

**Table 4. Adjusted share of OOPE on medicines out-off total household expenditure according to quintile of household expenditure.** HIES, Bangladesh, 2016/17.

| Threshold | Overall | Quintile of monthly household expenditure per equivalent adult | | | | |
|---|---|---|---|---|---|---|
| | | 1st | 2nd | 3th | 4th | 5th |
| ≥10% | 26.178 [25.003−27.353] | 27.279 [25.475−29.083] | 26.246 [24.583−27.909] | 26.442 [24.485−28.398] | 27.525 [25.524−29.527] | 23.614 [21.276−25.952] |
| ≥15% | 6.820 [6.362−7.278] | 6.649 [5.809−7.488] | 6.760 [5.991−7.529] | 7.200 [6.241−8.160] | 7.113 [6.255−7.972] | 6.396 [5.429−7.363] |
| ≥20% | 1.824 [1.619−2.028] | 1.746 [1.390−2.102] | 1.725 [1.370−2.080] | 1.911 [1.399−2.423] | 1.910 [1.488−2.333] | 1.825 [1.402−2.249] |
| ≥30% | 0.138 [0.095−0.181] | 0.124 [0.038−0.210] | 0.078 [0.007−0.149] | 0.200 [0.068−0.333] | 0.128 [0.046−0.210] | 0.159 [0.058−0.260] |

We use household capacity to pay to calculate the proportion of medicines expenditure out of total household expenditure. We find that 71% of households spent more than 10% of their entire disposable income on medicines (Table 5). Large disparities exist between the poorest and the wealthiest households. While 23.8% of the poorest households spent over 20% of their disposable household income on medicines, only 12.1% of the wealthiest households did so. The disparity between the lowest and wealthiest households increases with higher thresholds: while 0.5% of household spent over 40% of their disposable household income on medicines, only 0.2% of the wealthiest households did so.

## Discussion

Bangladesh has made remarkable progress in relation to its population health and economic development over the past 20 years [33]. Ensuring the continuation of this progress depends partly on strengthening existing social programs and developing new programs such as a financial health-related protection for all citizens [34]. To guide the development and implementation of these policies, it is critical to identify determinants of OOPE on healthcare,

**Table 5. Adjusted share of OOPE on medicines out-off household's capacity to pay according to quintile of household expenditure.** HIES, Bangladesh, 2016/17.

| Threshold | Overall | Quintile of monthly household expenditure per equivalent adult | | | | |
|---|---|---|---|---|---|---|
| | | 1st | 2nd | 3th | 4th | 5th |
| ≥10% | 71.058 [69.262−72.854] | 80.376 [78.767−81.984] | 78.532 [76.635−80.429] | 74.696 [72.274−77.118] | 69.612 [66.928−72.296] | 53.394 [49.288−57.500] |
| ≥15% | 42.213 [40.557−43.869] | 52.660 [50.628−54.691] | 48.027 [45.742−50.312] | 43.939 [41.495−46.383] | 40.161 [37.592−42.729] | 27.519 [24.531−30.508] |
| ≥20% | 18.547 [17.568−19.525] | 23.816 [22.170−25.463] | 20.960 [19.493−22.427] | 19.501 [17.748−21.253] | 16.957 [15.380−18.534] | 12.078 [10.518−13.637] |
| ≥30% | 2.760 [2.503−3.016] | 3.458 [2.796−4.120] | 3.114 [2.594−3.633] | 2.806 [2.197−3.414] | 2.613 [2.118−3.108] | 1.885 [1.460−2.310] |
| ≥40% | 0.296 [0.230−0.363] | 0.527 [0.329−0.725] | 0.268 [0.126−0.412] | 0.250 [0.099−0.401] | 0.250 [0.120−0.380] | 0.206 [0.092−0.320] |

especially those that determine OOPE on medicines, as these represent the largest proportion of healthcare OOPE.

The findings of this present study fill an important knowledge gap in terms of OOPE on medicines in Bangladesh. We show that the probability of having any healthcare expenditure within the previous month is high (74.4%)–in other words healthcare expenditures are frequent and medicines themselves represent nearly three quarters of all health related OOPE (71.5%). Both findings show the significance of medicines expenditure and the importance of including medicines in the benefit package of any Bangladeshfinancial protection program. Financial healthcare protection is much more relevant for poorer households as it significantly affects their overall household expenditure. Enrollment of poor households in social insurance programs is often a critical challenge to reach those most in need. For example, Seguro Popular, the pro-poor insurance program that Mexico implemented between 2003 and 2018, included medicines as part of this benefit catalogue after recognizing its relevance especially on poor households [35]. Moreover, the fact that almost all OOPE by households was for outpatient care highlights the critical importance of including outpatient medicines benefits in any effective financial protection scheme. In several countries, pharmaceutical benefit packages are limited to inpatient care which has resulted in insufficient financial protection of households [36].

Our findings not only show that poorer households pay proportionally much more for medicines than wealthier households, but also that the disparity between them is very large. When looking at medicines OOPE as a fraction of total household expenditure on healthcare, poor households pay nearly double that of their wealthiest counterparts. Household expenditure that exceeds 30% is regarded as impoverishing [37]. One of our most important findings is that 0.1%—one in every 1,000- Bangladeshi households spent more than 30% of their total monthly household expenditure on medicines. This means that medicines expenditure in Bangladesh can result in an estimated 376,000 households incurring catastrophic expenditure every month, and rural households are more affected compared to urban ones. The large number of households affected each month has the potential to significantly reduce opportunities for economic development and welfare of the population that is already very vulnerable.

Furthermore, our results demonstrate that chronic diseases have an increasing impact on household OOPE expenditure. Inclusion of common NCDs in an insurance benefit package is therefore critical to lower the probability, and especially the amount, of OOPE in NCDs. Cancer is particularly associated with very costly treatment and there is growing literature about what is called the "financial toxicity" of cancer treatment [37], especially in countries where the entire costs are born by patients, as it is in Bangladesh. Cost drivers are primarily the cost of surgery, radiation and medicines to treat cancer along with other indirect costs such as transportation, food, childcare [36]. Moreover, with the arrival of many biological oncology medicines, the costs of treatment have dramatically increased [38].

## Limitations

The results of this study have several limitations. First, this observational and cross-sectional analysis explores associations and does not prove causation. Second, we do not rule-out the existence of recall bias and lower accuracy in the self-report of analyzed variables. However, the HIES is the most detailed national representative survey available anywhere and we note that HIES uses a sophisticated method of collecting data to minimize recall bias. For instance, data pertaining to daily consumption is collected by the same enumerator every day visiting the household. A third limitation is that health expenditure is largely influenced by health status and the Bangladesh 2016–17 HIES provides only partial information on diagnoses or any other clinical information for household members. Fourth, we used the household as a unit of analysis. Although this is standard practice, it is noteworthy that it does not account for any complexity of diversity of families. Fifth, the data on health are self-reported which can introduce recall errors due to the fact that the respondent may not know or remember health related information. This could have affected our analysis of expenditure by type of disease. Sixth, this study does not analyze medicines prices or price elasticity as a factor affecting medicines OOPE because information on medicine prices is not collected as part of the survey. Linking outside data source with this analysis is very challenging as individuals in the survey do not report on specific products that they purchased. Finally, this study focuses on determinants of OOPE that are recorded within this survey. We did not link other databases to study determinants of expenditure such as distance to the nearest government/private hospital, nearest public/private clinics, nearest public/private dispensary, and availability of doctors, specialists, dentists per 1,000 of population as they might have an explanatory power.

## Conclusions

The introduction of efficient financial protection for the population is critical to protect households from increasing health and medicine expenditure. Our study shows that financial protection should be targeted to the poorest quintiles, enrollment of rural households should be ensured, and outpatient medicines benefits should including those for NCDs. The results of Bangladesh's 2016 HIES can serve as a baseline for measuring the progress achieved by the introduction of a new financial protection scheme in health with particular focus on medicines. Findings from this study would be also supportive to the healthcare financing strategy of the Bangladesh's Government for monitoring the progression towards UHC.

## Supporting information

**S1 File. Adjusted incremental risk of OOPE on medicines according to household presence chronic diseae.** HIES, Bangladesh, 2016.
(DOCX)

## Acknowledgments

The authors thank Kevin Gallagher for his comments on an earlier version of this manuscript.

## Author Contributions

**Conceptualization:** Warren A. Kaplan, Veronika J. Wirtz.

**Data curation:** Edson Serván-Mori, Md Deen Islam.

**Formal analysis:** Edson Serván-Mori, Md Deen Islam.

**Investigation:** Edson Serván-Mori, Veronika J. Wirtz.

**Methodology:** Edson Serván-Mori.

**Project administration:** Rachel Thrasher, Veronika J. Wirtz.

**Supervision:** Warren A. Kaplan, Rachel Thrasher, Veronika J. Wirtz.

**Validation:** Md Deen Islam.

**Writing – original draft:** Edson Serván-Mori, Warren A. Kaplan, Veronika J. Wirtz.

**Writing – review & editing:** Edson Serván-Mori, Md Deen Islam, Warren A. Kaplan, Rachel Thrasher, Veronika J. Wirtz.

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
