## [Decision Letter · Decision Letter 0]

23 Jun 2021

PONE-D-21-14254

Out-of-pocket expenditure on medicines in Bangladesh: an analysis of the national household income and expenditure survey 2016-17

PLOS ONE

Dear Dr. Wirtz,

Thank you for submitting your manuscript to PLOS ONE. After careful consideration, we feel that it has merit but does not fully meet PLOS ONE’s publication criteria as it currently stands. Therefore, we invite you to submit a revised version of the manuscript that addresses the points raised during the review process.

We look forward to receiving your revised manuscript.

Kind regards,

Mohammad Bellal Hossain

Academic Editor

PLOS ONE

Reviewers' comments:

Reviewer's Responses to Questions

**Comments to the Author**

1. Is the manuscript technically sound, and do the data support the conclusions?

Reviewer #1: Partly

Reviewer #2: Partly

2. Has the statistical analysis been performed appropriately and rigorously? 

Reviewer #1: No

Reviewer #2: Yes

3. Have the authors made all data underlying the findings in their manuscript fully available?

Reviewer #1: Yes

Reviewer #2: No

4. Is the manuscript presented in an intelligible fashion and written in standard English?

Reviewer #1: Yes

Reviewer #2: Yes

5. Review Comments to the Author

Reviewer #1: The paper examines the determinants of OOP expenses for medicines in Bangladesh using a set of cross-sectional data. The paper is well-written, but I have the following concerns on the current version of it:

1. The studies examining the determinants of OOP healthcare expenses in different contexts are voluminous. Contemporary researchers have now moved towards more experimental type research designs like randomized control trials to examine behavioral aspects of individual choices with regard to household expenses, and those studies still make significant contributions to the literature on consumer behavior. In this backdrop, this study uses a set of cross-sectional data from Bangladesh HIES 2016/2017 to examine the determinants and their association with OOP household expenses on medicines. I am therefore wondering whether the study makes any significant contribution to the literature. What is the novelty of the study? Which research gap the study is going to bridge? Very precisely, you need to elaborate what the contributions of this study are.

2. The study is lacking a sound theoretical foundation as well. You might want to postulate related hypotheses based on a theory or a set of theories as this study uses the deductive research approach. Therefore, my recommendation would be to develop a conceptual framework using related theories of consumer behavior before proposing the methodology.

3. Did you use the total sample or only the sample of households with positive expenditure for medicines to estimate fractional models and OLS regression? It is not clear from the result tables as they do not include the number of observations used for each model. First, I recommend authors to include vital information like number of observations, post-estimation test results, and measures of model fit under each model. Also, as I understood, household decision making process with regard to the demand for medicines has two stages: Whether to spend OOP for medicines or not, if yes, then how much to be spent. The current analytical process has not taken into account this two-stage nature of household decision making. For instance, Tobit model, Double hurdle model, and Heckman two-stage model may be better alternatives to check the robustness of current findings and to account for approximately the real nature of household decision making process for the demand for medicines.

4. Household expenses for healthcare should be examined on a broader backdrop of healthcare utilization. The magnitude of OOP expenses for medicines depends on whether a household utilizes in-patient care, out-patient care, or clinic services like dental clinics and maternal clinics (Kumara and Samaratunge, 2019). Otherwise, the analysis on OOP expenses would be incomplete. Can you statistically investigate the interplay between OOP expenses for medicines and healthcare utilization? If required, use the following study for more literature

Ajantha Sisira Kumara and Ramanie Samaratunge (2019), Relationship between healthcare utilization and household out-of-pocket healthcare expenditure: Evidence from an emerging economy with a free healthcare policy, Social Science & Medicine 235, pp. 1-12

5. The importance of supply-side factors in determining OOP expenses for medicines is completely ignored from the study, making it lopsided. Do you have data in HIES on healthcare supply-side factors? For instance, distance to the nearest government /private hospital, nearest public/private clinics, nearest public/private dispensary, and availability of doctors, specialists, dentists per 1,000 of population as they might have an explanatory power. My recommendation would be incorporate such variables and see whether they play a role in Bangladesh like in other contexts.

6. As the study uses a set of cross-sectional data, you need to be careful of addressing the issue of endogeneity. As the study has many omitted variables, it would definitely lead the way for the issue. For instance, you have omitted the variable of health insurance ownership which might affect both OOP healthcare expenses and health status of household members (via moral hazard), leading the way for endogeneity. Thus, checking the results for endogeneity would be advised, and endogeneity-corrected models like IV-regression models might need to be applied to have more accurate results.

Reviewer #2: I have some major concern on conceptualization and analyses of the paper.

Conceptualization: The paper analyzed OOP on medicine in Bangladesh which is partial. as authors mentioned medicine accounts 61% of OOP in the country and so it is a unique analyses. I differ in this ground. The paper would have focused on OOPE and a sub-section on medicine. If such analyses have already been carried out, authors need to think differently.

Methods: Authors used budget share approach and provided the varying incidence of CHE. Findings suggest, at 10% threshold, the CHE of fourth quintile is higher than first quintile. This is primarily due to limitations of budget share approach in estimating the CHE. I suggest the authors should use the capacity to pay approach as the expd and income survey provides required variable for estimation of CHE.

In table 4, authors must mention whether it is incidence of CHE ?

Adjusted per equivalent adult and out of those households

192 who spent on health, their median total monthly health-related OOPE was US$3.14.

I am not sure whether is is per capita or per household. I think it should be per capita

authors must mention how they derive adult equivalent scale ? What weight they assign to children? Need mentioning

Authors disease classification is not adequate. Moreover, they must mention as limitation of self reported diseases as expd survey typically collect self reported data

Abstract: Background and objective

There is only one sentence of aim

Keep a line on background

6. PLOS authors have the option to publish the peer review history of their article (what does this mean?). If published, this will include your full peer review and any attached files.

Reviewer #1: **Yes: **Ajantha Sisira Kumara

Reviewer #2: **Yes: **Sanjay K Mohanty

---

## [Author Response · Author response to Decision Letter 0]

13 Oct 2021

Responses to the reviewers' comments

Manuscript [PONE-D-21-14254] - [EMID:661117b46576190d]

We would like to thank the reviewer for the constructive comments. We have carefully addressed the each of the reviewer’s comments in this revision. We believe that after incorporating your thoughtful feedback, our manuscript has been greatly improved.

Reviewer #1:

The paper examines the determinants of OOP expenses for medicines in Bangladesh using a set of cross-sectional data. The paper is well-written, but I have the following concerns on the current version of it: 1. The studies examining the determinants of OOP healthcare expenses in different contexts are voluminous. Contemporary researchers have now moved towards more experimental type research designs like randomized control trials to examine behavioral aspects of individual choices with regard to household expenses, and those studies still make significant contributions to the literature on consumer behavior. In this backdrop, this study uses a set of cross-sectional data from Bangladesh HIES 2016/2017 to examine the determinants and their association with OOP household expenses on medicines. I am therefore wondering whether the study makes any significant contribution to the literature. What is the novelty of the study? Which research gap the study is going to bridge? Very precisely, you need to elaborate what the contributions of this study are.

RESPONSE: We agree with the reviewer that different designs are used to study out-of-pocket expenditure. Our original contribution lays in using a secondary dataset (the HIES household survey) that has not be explored to answer an important policy question: what factors contribute to high OOP on medicines in Bangladesh in 2016? As we mention in our introduction there are other studies which have looked at OOP on health in general. However, no recent study has analyzed the factors that drive high OOP on medicines in Bangladesh, a country with rapidly increasing life expectancy fueling the epidemiological transition from a largely infectious towards a growing non-communicable disease burden. 

We have changed the background section to address the point raised by the reviewer: “Although there are some previous studies of the OOPE on medicines in Bangladesh the contributions of this study are twofold: there is a gap in our understanding of the determinants of OOPE on medicines. Knowing these determinants would support the development of policies to protect households from financial hardship. Additionally, this study uses the most recent national-level survey data on healthcare utilization and OOP expenditure for out-patients in Bangladesh making the study highly relevant from a public policy standpoint.”

2. The study is lacking a sound theoretical foundation as well. You might want to postulate related hypotheses based on a theory or a set of theories as this study uses the deductive research approach. Therefore, my recommendation would be to develop a conceptual framework using related theories of consumer behavior before proposing the methodology.

RESPONSE: Thank you for your suggestion to add more on the theoretical foundation in the background of our manuscript. We have added the following section:

“Out-of-pocket expenditure is amendable by public policy: for instance, the implementation of health insurance should protect households from large out-of-pocket expenditure on health, including medicines. Furthermore, policies regulating the payment of providers also have an influence on their behaviors, e.g., the ordering of diagnostic tests, the prescribing of medicines and recommendations of surgery and other types of treatment. This makes the study of out-of-pocket expenditures relevant for setting health policies and assessing their effect on economic development and poverty reduction in a country. There is ample literature on the study of OOPE [8], the methodological foundation [9] and its current application to assess progress on UHC [1].” It is noteworthy that this study is not analyzing consumer behavior.

3. Did you use the total sample or only the sample of households with positive expenditure for medicines to estimate fractional models and OLS regression? It is not clear from the result tables as they do not include the number of observations used for each model. First, I recommend authors to include vital information like number of observations, post-estimation test results, and measures of model fit under each model. Also, as I understood, household decision making process with regard to the demand for medicines has two stages: Whether to spend OOP for medicines or not, if yes, then how much to be spent. The current analytical process has not taken into account this two-stage nature of household decision making. For instance, Tobit model, Double hurdle model, and Heckman two-stage model may be better alternatives to check the robustness of current findings and to account for approximately the real nature of household decision making process for the demand for medicines.

RESPONSE: Thank you for these reflections. For each of the tables we have included the total number of observations and included an explanation whether the analysis included the total sample or only a subgroup such as those households with a positive expenditure on health:

We now have used the Heckman two-stage model for our analysis to consider the presence of the selection bias that the reviewer mentions:

“Following previous studies [19], our estimations had to consider the presence of a selection bias related to the decision to spend funds on health because there may be particular household characteristics that increase the probability of health expenditure. This bias is applied to all households. Following Heckman (1979) [28], we estimated a logistic model to calculate the conditional probability that a given household would record any given health OOPE (as a function of the household characteristics mentioned before). We then calculated the Mills ratio [29] to capture the magnitude of the selection bias for each household analyzed. Subsequently, this parameter was incorporated as a regressor in the four regression models described above.”

It should be noted that we do not study causal inference. We are studying the magnitude of out-of-pocket expenditure and the factors that are associated with the magnitude and frequency of medicines expenditure.

4. Household expenses for healthcare should be examined on a broader backdrop of healthcare utilization. The magnitude of OOP expenses for medicines depends on whether a household utilizes in-patient care, out-patient care, or clinic services like dental clinics and maternal clinics (Kumara and Samaratunge, 2019). Otherwise, the analysis on OOP expenses would be incomplete. Can you statistically investigate the interplay between OOP expenses for medicines and healthcare utilization? If required, use the following study for more literature Ajantha Sisira Kumara and Ramanie Samaratunge (2019), Relationship between healthcare utilization and household out-of-pocket healthcare expenditure: Evidence from an emerging economy with a free healthcare policy, Social Science & Medicine 235, pp. 1-12

RESPONSE: We would like to thank the reviewer for the recommendation of the publication. We have now added health care utilization as one of the factors that influences health expenditure. We have revised all tables to reflect this change as well as the methods and results section to accommodate the incorporation of health service use as a factor influencing health and medicine expenditures. As expected, health service use increases health and medicines expenditure and is an important driver not only of the probability of expenditure but also the amount of spending.

5. The importance of supply-side factors in determining OOP expenses for medicines is completely ignored from the study, making it lopsided. Do you have data in HIES on healthcare supply-side factors? For instance, distance to the nearest government /private hospital, nearest public/private clinics, nearest public/private dispensary, and availability of doctors, specialists, dentists per 1,000 of population as they might have an explanatory power. My recommendation would be incorporate such variables and see whether they play a role in Bangladesh like in other contexts.

RESPONSE: The purpose of this study is to analyze household and individual factors that are associated with medicines OOPE. The reviewer is correct that additional data on the supply side could be added to complete this analysis. However, the survey does not provide geospatial data at the unit of analysis which is the household. Adding supply side factors such as nearest public dispensary, etc. would be approximations based on the smallest geographical area. We have added in the discussion a section where we expand on the need for analysis of the supply-side factors.

“Finally, this study focuses on determinants of out-of-pocket expenditure that are recorded within this survey. We did not link other databases to study determinants of expenditure that are not included in the survey such as distance to the nearest government /private hospital, nearest public/private clinics, nearest public/private dispensary, and availability of doctors, specialists, dentists per 1,000 of population as they might have an explanatory power.”

6. As the study uses a set of cross-sectional data, you need to be careful of addressing the issue of endogeneity. As the study has many omitted variables, it would definitely lead the way for the issue. For instance, you have omitted the variable of health insurance ownership which might affect both OOP healthcare expenses and health status of household members (via moral hazard), leading the way for endogeneity. Thus, checking the results for endogeneity would be advised, and endogeneity-corrected models like IV-regression models might need to be applied to have more accurate results.

RESPONSE: The reviewer is correct in asserting that health insurance would be expected to be an important determinant of out-of-pocket expenditure. However, in Bangladesh it is estimated that less than 1% of the population have health insurance (Rahman, 2019). Furthermore, the survey instrument does not have a separate question on health insurance affiliation only whether the household is Beneficiary/member of any safety nets/social program. We have included this variable in our models: “Beneficiary/member of any safety nets/social program”. Our study results show that only one fifth (21.18%) of the population are covered by some form of safety nets or social program which include health insurance. The regression models show that being covered by a safety nets/social program reduces the share of OOP on medicines out of total health expenditure, reduces the amount of OOP on medicines and the share of OOPE on medicines out of household expenditure.

We have added this aspect in the results section: “Being covered by a safety nets/social program reduces the share of OOP on medicines out of total health expenditure, reduces the amount of OOP on medicines and the share of OOPE on medicines out of household expenditure.” However, it is important to recognize that the survey manual clarifies that the social programs do not include support of health service use or reimbursement of medical expenditure.

Reviewer #2:

1. I have some major concern on conceptualization and analyses of the paper. Conceptualization: The paper analyzed OOP on medicine in Bangladesh which is partial as authors mentioned medicine accounts 61% of OOP in the country and so it is a unique analyses. I differ in this ground. The paper would have focused on OOPE and a sub-section on medicine. If such analyses have already been carried out, authors need to think differently.

RESPONSE: The reviewer is correct that OOPE on medicines is only a part of the total OOP on health. We expanded the background section to explain the relevance of medicine OOP in the context of health expenditure and its relation to total household expenditure.

“Out of pocket expenditure is amendable by public policy: for instance, the implementation of health insurance should protect households from large out-of-pocket expenditure on health including medicines. Furthermore, policies regulating the payment of providers also have an influence on their behavior to order diagnostic tests, prescribe medicines and recommend surgery and other types of treatment. This makes the study of out-of-pocket expenditure relevant for setting health policies and assessing their effect on economic development and poverty reduction in a country. There is ample literature on the study of out-of-pocket expenditure (Wagstaff and van Doorlaer, 2003), the methodological foundation (Xu et al, 2007) and its current application to assess progress on UHC (Wagstaff et al, 2019).

More specifically, the study of medicines OOP within health OOP expenditure is very relevant as evidence shows that in many countries the majority of health OOP is for medicines: for instance, data from Southeast Asia show that in India 90% of out of pocket expenditure on health is on medicines, in Nepal it is 88%, and Indonesia it is 78%. Not only in relation to health OOP are medicines OOP expenditure relevant; they are also important as a proportion of the total household expenditure: according to the World Health Survey, up to 9·5% of the total expenditure of poorer households in LMICs is spent on medicines, far higher than the 3·5% expended by poorer households in high income countries (HICs) (Wagner et al, Health Policy). Approximately half (41%-56%) of households spent 100% of their health care expenses on medicines (Wagner et al, Health Policy).

Since medicines OOP is a larger driver of the overall health OOP and countries have made a commitment to Universal Health Coverage, studies focusing on medicines OOP become increasingly relevant.”

As we explain, focusing the manuscript on medicines OOP is important. At the same time, our manuscript reports on health OOP. While medicines OOP is the central theme of the manuscript, we believe it gives sufficient attention to health OOP as well.

2. Methods: Authors used budget share approach and provided the varying incidence of CHE. Findings suggest, at 10% threshold, the CHE of fourth quintile is higher than first quintile. This is primarily due to limitations of budget share approach in estimating the CHE. I suggest the authors should use the capacity to pay approach as the expd and income survey provides required variable for estimation of CHE. In table 4, authors must mention whether it is incidence of CHE?

RESPONSE: Thank you for this suggestion. We have now revised the analysis and incorporated the reviewer’s suggestion to include the capacity to pay approach. We have also added an analysis that uses the disposable household income as described by Xu et al. 2007. The methods section under the main outcome variables reads: “4) based in Wagstaff’s approach [2,8], the share of OOPE on medicines out of total household expenditure (reported as a ratio between zero and one), and 5) based in Xu’s approach [9,21], the share of OOPE on medicines out-off household’s capacity to pay (reported also as a ratio between zero and one).”

As expected the disparity between households in the poorest and the wealthiest quintile is even larger when using the capacity to pay approach regarding the proportion of medicine expenditure out of total household expenditure. The results section now reads: “Expressed as the capacity of the household to pay, the disparity between households in the poorest and the wealthiest quintile is even larger: while poor households spent nearly a one-fifth (19.85%) of their disposable income on medicines, the wealthiest households spent 7.6% on medicines.”

“Using the capacity to pay approach to calculate the proportion of medicines out of total household expenditure, we show that 71% of households spent more than 10% of their disposable income on medicines. Large disparities exist between the poorest and the wealthiest households: while 23.8% of the poorest households spent over 20% of their disposable household income on medicines, only 12.1% of the wealthiest households do so.”

We believe that following the reviewer’s suggestion and complementing our analysis using these two approaches have substantially strengthened the paper.

3. Adjusted per equivalent adult and out of those households 192 who spent on health, their median total monthly health-related OOPE was US$3.14. I am not sure whether is is per capita or per household. I think it should be per capita. Authors must mention how they derive adult equivalent scale? What weight they assign to children? Need mentioning

RESPONSE: Thank you for the question by the reviewer. We agree that it is important to be clear on the meaning of “adult equivalent” and how it was calculated: 

“Adult equivalent’ adjusts for the economy of scale in consumption – a household with three members, including children, for example, does not consume three times that of a one-person household. The equivalence scale considers the age of the household members and establishes a standardization that allows comparison.”

Per ‘Adult equivalent’ adjustment is standard in welfare studies. It has been developed to account for the variation in the consumption of goods by household members on the basis of age and economies of scale of consumption. For more information we refer the reviewer to the following resource which we have cited in the paper: Haughton J, Khandker SR. Handbook on Poverty and Inequality. 1st ed. World Bank Publications; 2009.

4. Authors disease classification is not adequate. Moreover, they must mention as limitation of self-reported diseases as expd survey typically collect self-reported data

RESPONSE: We thank the reviewer for this note. We have added to the limitation the following sentence: “Fifth, the data on health are self-reported which can introduce errors due to the fact that the respondent may not know or remember health related information. This could have affected our analysis of expenditure by type of disease.”

5. Abstract: Background and objective. There is only one sentence of aim. Keep a line on background

RESPONSE: We have added a line “High OOPE increases the probability that households will become impoverished or will forgo needed care.”

REFERENCE

Rahman, S. Universal health coverage in Bangladesh: the challenges. The Financial Express 2021, Sep 2. https://thefinancialexpress.com.bd/views/universal-health-coverage-in-bangladesh-the-challenges-1549037007 [accessed Sep 1 2021]

According to the World Health Survey, up to 9·5% of the total expenditure of poorer households in LMICs is spent on medicines, far higher than the 3·5% expended by poorer households in high income countries (HICs). Approximately half (41%-56%) of households spent 100% of their health care expenses on medicines (Wagner et al, Health Policy).

---

## [Decision Letter · Decision Letter 1]

17 Dec 2021

PONE-D-21-14254R1Out-of-pocket expenditure on medicines in Bangladesh: an analysis of the national household income and expenditure survey 2016-17PLOS ONE

Dear Dr. Wirtz,

Thank you for submitting your manuscript to PLOS ONE. After careful consideration, we feel that it has merit but does not fully meet PLOS ONE’s publication criteria as it currently stands. Therefore, we invite you to submit a revised version of the manuscript that addresses the points raised during the review process. Please submit your revised manuscript by Jan 31 2022 11:59PM. If you will need more time than this to complete your revisions, please reply to this message or contact the journal office at plosone@plos.org. Please include the following items when submitting your revised manuscript:A rebuttal letter that responds to each point raised by the academic editor and reviewer(s). You should upload this letter as a separate file labeled 'Response to Reviewers'.A marked-up copy of your manuscript that highlights changes made to the original version. You should upload this as a separate file labeled 'Revised Manuscript with Track Changes'.An unmarked version of your revised paper without tracked changes. You should upload this as a separate file labeled 'Manuscript'.If applicable, we recommend that you deposit your laboratory protocols in protocols.io to enhance the reproducibility of your results. Protocols.io assigns your protocol its own identifier (DOI) so that it can be cited independently in the future. For instructions see: https://journals.plos.org/plosone/s/submission-guidelines#loc-laboratory-protocols. Additionally, PLOS ONE offers an option for publishing peer-reviewed Lab Protocol articles, which describe protocols hosted on protocols.io. Read more information on sharing protocols at https://plos.org/protocols?utm_medium=editorial-email&utm_source=authorletters&utm_campaign=protocols.

We look forward to receiving your revised manuscript.

Kind regards,

Mohammad Bellal Hossain

Academic Editor

PLOS ONE

Reviewers' comments:

Reviewer's Responses to Questions

**Comments to the Author**

1. If the authors have adequately addressed your comments raised in a previous round of review and you feel that this manuscript is now acceptable for publication, you may indicate that here to bypass the “Comments to the Author” section, enter your conflict of interest statement in the “Confidential to Editor” section, and submit your "Accept" recommendation.

Reviewer #1: All comments have been addressed

Reviewer #2: All comments have been addressed

2. Is the manuscript technically sound, and do the data support the conclusions?

Reviewer #1: Partly

Reviewer #2: Partly

3. Has the statistical analysis been performed appropriately and rigorously? 

Reviewer #1: Yes

Reviewer #2: No

4. Have the authors made all data underlying the findings in their manuscript fully available?

Reviewer #1: Yes

Reviewer #2: Yes

5. Is the manuscript presented in an intelligible fashion and written in standard English?

Reviewer #1: Yes

Reviewer #2: Yes

6. Review Comments to the Author

Reviewer #1: I am satisfied with the revisions. However, the paper needs copy-editing before publishing. Also, all tables should be formatted professionally.

Reviewer #2: While you have estimated the catastrophic health spending using capacity to pay approach, you have not estimated the estimates using threshold of 40%. In CTP approach a single threshold of 40% is used. If you will do so, you will find difefrent estimates by quintile. Your revised table 4 is not correct. It is based on budget share approach Pl read the following reference to get complete idea of estimating CHE using CTP approach

https://equityhealthj.biomedcentral.com/articles/10.1186/s12939-021-01421-6

Addressing data and methodological limitations in estimating catastrophic health spending and impoverishment in India, 2004–18

7. PLOS authors have the option to publish the peer review history of their article (what does this mean?). If published, this will include your full peer review and any attached files.

Reviewer #1: **Yes: **Ajantha Sisira Kumara

Reviewer #2: **Yes: **Sanjay K Mohanty

---

## [Author Response · Author response to Decision Letter 1]

8 Feb 2022

We would like to thank the reviewer for the constructive comments. We have carefully addressed the each of the reviewer’s comments in this revision. We believe that after incorporating your thoughtful feedback, our manuscript has been greatly improved. Our responses to the reviewer’s comment can be found in bold. 

Reviewer #2: While you have estimated the catastrophic health spending using capacity to pay approach, you have not estimated the estimates using threshold of 40%. In CTP approach a single threshold of 40% is used. If you will do so, you will find different estimates by quintile. Your revised table 4 is not correct. It is based on budget share approach Pl read the following reference to get complete idea of estimating CHE using CTP approach

https://equityhealthj.biomedcentral.com/articles/10.1186/s12939-021-01421-6

Addressing data and methodological limitations in estimating catastrophic health spending and impoverishment in India, 2004–18

Authors’ response: Thank you very much for your comment. In response to your suggestion and following the article you shared with us, we disaggregated Table 4 into two separate tables (see below), the first (Table 4) for the adjusted ratio of spending on medicines with respect to total household spending, and the second (Table 5) for the adjusted ratio of spending on medicines with respect to of households' ability to pay. In Table 5 we have the additional threshold of 40% of the households' ability to pay. We clarify this point in the methods section as follows:

[…] Finally, based on the first regression analysis results, we estimated the adjusted share of OOPE on medicines out of total household expenditure (considering the following thresholds of the total household expenditure: 10, 15, 20, 30%); and out of a household’s capacity to pay by considering different thresholds (10, 15, 20, 30 and 40% of the total household expenditure) [31] and according to the quintile of household expenditure per equivalent adult […]

We have added further descriptions of the results: 

“The disparity between the lowest and wealthiest households increases with higher thresholds: while 0.5% of household spent over 40% of their disponible household income on medicines, only 0.2% of the wealthiest households do so.” 

Reference: 31. Mohanty, S.K., Dwivedi, L.K. Addressing data and methodological limitations in estimating catastrophic health spending and impoverishment in India, 2004–18. Int J Equity Health 20, 85 (2021). https://doi.org/10.1186/s12939-021-01421-6

We have included the revised Table 4 and a new Table 5 in the manuscript.

---

## [Decision Letter · Decision Letter 2]

22 Apr 2022

PONE-D-21-14254R2Out-of-pocket expenditure on medicines in Bangladesh: an analysis of the national household income and expenditure survey 2016-17PLOS ONE

Dear Dr. Wirtz,

Thank you for submitting your manuscript to PLOS ONE. After careful consideration, we feel that it has merit but does not fully meet PLOS ONE’s publication criteria as it currently stands. Therefore, we invite you to submit a revised version of the manuscript that addresses the points raised during the review process.

We look forward to receiving your revised manuscript.

Kind regards,

Mohammad Bellal Hossain

Academic Editor

PLOS ONE

Journal Requirements:

Reviewers' comments:

Reviewer's Responses to Questions

**Comments to the Author**

1. If the authors have adequately addressed your comments raised in a previous round of review and you feel that this manuscript is now acceptable for publication, you may indicate that here to bypass the “Comments to the Author” section, enter your conflict of interest statement in the “Confidential to Editor” section, and submit your "Accept" recommendation.

Reviewer #1: All comments have been addressed

Reviewer #2: All comments have been addressed

2. Is the manuscript technically sound, and do the data support the conclusions?

Reviewer #1: Partly

Reviewer #2: Yes

3. Has the statistical analysis been performed appropriately and rigorously? 

Reviewer #1: Yes

Reviewer #2: Yes

4. Have the authors made all data underlying the findings in their manuscript fully available?

Reviewer #1: Yes

Reviewer #2: Yes

5. Is the manuscript presented in an intelligible fashion and written in standard English?

Reviewer #1: Yes

Reviewer #2: Yes

6. Review Comments to the Author

Reviewer #1: The paper has been written well, and it analyzes the patterns, trends, and determinants of OOP expenses for medicines in Bangladeshi households using nation-wide survey data from the HIES. The paper provides a lot of empirical evidence on the subject using appropriate analytical methods. For instance, the paper analyses trends and determinants of OOP household expenses on medicines across different income quintiles. Please check whether you could attend the followings which may further improve your paper:

• From the view point of public policy, it may be important to discuss about income elasticity of OOP expenses on medicine. Can you calculate the elasticity using double-log regression framework? Analyzing the data from elasticity ground may be more important than analyzing the same across different income quintiles. This type of an analysis may be useful to determine whether medicine is an essential or a luxury for Bangladeshi households.

• The price-levels of medicines play a significant role in determining the extent of OOP expenses on medicines. Did take price-factor into account when analyzing? If not, the current analysis is lop-sided, and resultantly, the paper is missing a big picture coming from the price factor. You may use the data from price indices of medicine or any other source to capture the impact of prices on OOP expenses on medicines.

• The covariates that the paper considered represent only the demand-side of medicines. The paper misses the story coming from supply-side factors in determining OOP expenses on medicines. Availability of public/private avenues for treatments, proximity to such avenues, health insurance coverage, facilities available in public/private avenues, influence of doctors are some of supply-side factors. Can you strengthen the analysis by incorporating the information pertaining to those supply-side factors? Accordingly, you may strengthen the sections of discussions and conclusions by describing the situation and the role of such supply-side factors in determining OOP expenses on medicines.

Reviewer #2: Thank you for implementing necessary changes. the paper has improved in content and presentation. Limitations of the study may be highlighted

7. PLOS authors have the option to publish the peer review history of their article (what does this mean?). If published, this will include your full peer review and any attached files.

Reviewer #1: **Yes: **Ajantha Sisira Kumara

Reviewer #2: **Yes: **Sanjay K Mohanty

---

## [Author Response · Author response to Decision Letter 2]

20 May 2022

Reviewer #1: The paper has been written well, and it analyzes the patterns, trends, and determinants of OOP expenses for medicines in Bangladeshi households using nation-wide survey data from the HIES. The paper provides a lot of empirical evidence on the subject using appropriate analytical methods. For instance, the paper analyses trends and determinants of OOP household expenses on medicines across different income quintiles. 

RESPONSE: Thank you. 

Please check whether you could attend the followings which may further improve your paper:

• From the view point of public policy, it may be important to discuss about income elasticity of OOP expenses on medicine. Can you calculate the elasticity using double-log regression framework? Analyzing the data from elasticity ground may be more important than analyzing the same across different income quintiles. This type of an analysis may be useful to determine whether medicine is an essential or a luxury for Bangladeshi households.

RESPONSE: As the reviewer suggests we have incorporated this aspect in the discussion section. 

“Sixth, this study does not analyze medicines prices or price elasticity as a factor affecting medicines out-of-pocket expenditure because information on medicine prices is not collected as part of the survey. Linking outside data source with the data set used in this study is challenging as individuals in the survey do not report on specific products that they purchased.”

Moreover, it is noteworthy to mention to the reviewer that medicine price information collected through market studies often do not reflect the prices patients pay for their medicines.

• The price-levels of medicines play a significant role in determining the extent of OOP expenses on medicines. Did take price-factor into account when analyzing? If not, the current analysis is lop-sided, and resultantly, the paper is missing a big picture coming from the price factor. You may use the data from price indices of medicine or any other source to capture the impact of prices on OOP expenses on medicines.

RESPONSE: Thank you for this comment. We have included an explanation in the discussion section about this point. See response to the comment above. 

It is important to note that the National Household and Expenditure Survey asks the respondents to provide the information on the total costs of their medicines over the past 30 days. It does not ask the respondents about the price of each of the medicine purchased. We do not know how many medicines individuals consumed. 

It is also important to take into consideration that the amounts analyzed are at current market prices and that there are no reliable price indices for medicines in the different geographical areas, so the price factor mentioned could not be effectively controlled. Furthermore, it is important to mention that correcting for inflation would not modify the results obtained since this survey is cross-sectional and not a comparative one of different points of time.

• The covariates that the paper considered represent only the demand-side of medicines. The paper misses the story coming from supply-side factors in determining OOP expenses on medicines. Availability of public/private avenues for treatments, proximity to such avenues, health insurance coverage, facilities available in public/private avenues, influence of doctors are some of supply-side factors. Can you strengthen the analysis by incorporating the information pertaining to those supply-side factors? Accordingly, you may strengthen the sections of discussions and conclusions by describing the situation and the role of such supply-side factors in determining OOP expenses on medicines.

RESPONSE: The reviewer is correct that this analysis focuses on the demand-side. It would be very interesting to link the demand side data with the supply side. However, this is outside the scope of this manuscript. In addition, it is noteworthy, that linking supply side factors such as availability of medicines in public and private sector and prescriber behavior in the catchment area of individual households are often absent or not updated. We have included this limitation in the discussion sector. 

“Finally, this study focuses on determinants of out-of-pocket expenditure that are recorded within this survey. We did not link other databases to study determinants of expenditure such as distance to the nearest government/private hospital, nearest public/private clinics, nearest public/private dispensary, and availability of doctors, specialists, dentists per 1,000 of population as they might have an explanatory power.”

Reviewer #2: Thank you for implementing necessary changes. the paper has improved in content and presentation. Limitations of the study may be highlighted.

RESPONSE: Thank you. 

---

## [Decision Letter · Decision Letter 3]

13 Jul 2022

PONE-D-21-14254R3Out-of-pocket expenditure on medicines in Bangladesh: an analysis of the national household income and expenditure survey 2016-17PLOS ONE

Dear Dr. Wirtz,

Thank you for submitting your manuscript to PLOS ONE. After careful consideration, we feel that it has merit but does not fully meet PLOS ONE’s publication criteria as it currently stands. Therefore, we invite you to submit a revised version of the manuscript that addresses the points raised during the review process.

We look forward to receiving your revised manuscript.

Kind regards,

Mohammad Bellal Hossain

Academic Editor

PLOS ONE

Journal Requirements:

Reviewers' comments:

Reviewer's Responses to Questions

**Comments to the Author**

1. If the authors have adequately addressed your comments raised in a previous round of review and you feel that this manuscript is now acceptable for publication, you may indicate that here to bypass the “Comments to the Author” section, enter your conflict of interest statement in the “Confidential to Editor” section, and submit your "Accept" recommendation.

Reviewer #1: (No Response)

Reviewer #2: (No Response)

2. Is the manuscript technically sound, and do the data support the conclusions?

Reviewer #1: Partly

Reviewer #2: (No Response)

3. Has the statistical analysis been performed appropriately and rigorously? 

Reviewer #1: Yes

Reviewer #2: (No Response)

4. Have the authors made all data underlying the findings in their manuscript fully available?

Reviewer #1: Yes

Reviewer #2: (No Response)

5. Is the manuscript presented in an intelligible fashion and written in standard English?

Reviewer #1: Yes

Reviewer #2: (No Response)

6. Review Comments to the Author

Reviewer #1: Thank you for answering my previous review questions

However, I still believe that you can calculate income elasticity of medical expenditure at the household-level. For this purpose, you no need to have the data pertaining to unit prices of medicine. I hope your dataset has household income or otherwise, it can be proxied by household expenditure.

Further, I still believe that you can incorporate health-related supply-side factors to calculate their impact on OOP expenditure on medicine. Those data are available at the country's macro-level and can be collected from annual reports of relevant ministries.

I firmly believe that the paper will be in a better shape after you address at least these two issues. Congratulations!

Minor: Language editing is advisable.

Reviewer #2: (No Response)

7. PLOS authors have the option to publish the peer review history of their article (what does this mean?). If published, this will include your full peer review and any attached files.

Reviewer #1: **Yes: **Ajantha Sisira Kumara

Reviewer #2: **Yes: **Sanjay Mohanty

---

## [Author Response · Author response to Decision Letter 3]

7 Aug 2022

Dear Editor,

We would like to thank you for your response to our letter to the Editor on July 26 (see the Annex to this letter). You asked us to edit the language. In response to this request, we have significantly edited the language. You will see these changes in the track-change version that we uploaded.

Additional responses to the reviewer’s comments can be found in our letter to the Editor on July 15, 2022 (also in the Annex). 

Annex: Recent correspondence with PLoS

From: plosone <plosone@plos.org> 

Sent: Tuesday, July 26, 2022 5:49 PM

To: Wirtz, Veronika <vwirtz@bu.edu>

Subject: RE: PLOS ONE Decision: Revision required [PONE-D-21-14254R3]

Dear Dr. Wirtz,

Thank you for your patience, here is the response of the AE regarding to your concern in reviewers comment.

******

Thanks for your email. I have gone through the comments and responses. I think the authors are valid in their argument. But they still need to edit the language. Please ask them to do so, and then I will provide my final decision.

*******

Feel free to message me if you have any further question or concern

Kind regards,

Liezl Callo

Straive Editorial Assistant

PLOS ONE | plosone@plos.org

Boston, July 15,2022

Dear Editor

We are responding to the most recent request for changes to this manuscript. Respectfully, we would like to ask that the Journal make a final determination as to whether or not our manuscript [PONE-D-21-14254R3] - [EMID:e4ab9e939629ff6c] is acceptable for publication without making any additional changes. One of the reviewers, Ajantha Sisira Kumara, keeps insisting (now for the third time) that we provide two additional analyses. In our view, these analyses are both out of the scope of our study and, in fact, are not feasible. We have explained our reasoning in our detailed responses over the past three re-submissions, which can be found below. Indeed, we believe these requests would amount to writing a separate paper.

It is noteworthy that all other reviewers over the course of the three revisions have been satisfied with our responses and how we incorporated their comments except Ajantha Sisira Kumara who keeps coming back with the same request over and over regardless of our explanation.

Since it is clear from Kumara’s comments that nothing is wrong with the current manuscript and that this request for additional analyses is complementary and not essential, we respectfully ask the Journal to please accept the manuscript at this stage.

Sincerely,

Veronika J. Wirtz, B.Pharm., M.Sc., Ph.D., FISPE

Professor - Department of Global Health, Boston University School of Public Health 

Director, World Health Organization Collaborating Center in Pharmaceutical Policy 

vwirtz@bu.edu; T: +1 617 358 3046

https://www.bu.edu/sph/profile/veronika-wirtz/

pronouns: she/her/hers @VeroWirtz

1st Revision October 2021

Ajantha Sisira Kumara: “The importance of supply-side factors in determining OOP expenses for medicines is completely ignored from the study, making it lopsided. Do you have data in HIES on healthcare supply-side factors? For instance, distance to the nearest government /private hospital, nearest public/private clinics, nearest public/private dispensary, and availability of doctors, specialists, dentists per 1,000 of population as they might have an explanatory power. My recommendation would be incorporate such variables and see whether they play a role in Bangladesh like in other contexts.”

RESPONSE: The purpose of this study is to analyze household and individual factors that are associated with medicines OOPE. The reviewer is correct that additional data on the supply side could be added to complete this analysis. However, the survey does not provide geospatial data at the unit of analysis which is the household. Adding supply side factors such as nearest public dispensary, etc. would be approximations based on the smallest geographical area. We have added in the discussion a section where we expand on the need for analysis of the supply-side factors.

“Finally, this study focuses on determinants of out-of-pocket expenditure that are recorded within this survey. We did not link other databases to study determinants of expenditure that are not included in the survey such as distance to the nearest government /private hospital, nearest public/private clinics, nearest public/private dispensary, and availability of doctors, specialists, dentists per 1,000 of population as they might have an explanatory power.”

2nd Revision in December 2022

All comments successfully responded to. 

3rd Revision May 2022

Ajantha Sisira Kumara: “Please check whether you could attend the followings which may further improve your paper: • From the viewpoint of public policy, it may be important to discuss about income elasticity of OOP expenses on medicine. Can you calculate the elasticity using double-log regression framework? Analyzing the data from elasticity ground may be more important than analyzing the same across different income quintiles. This type of an analysis may be useful to determine whether medicine is an essential or a luxury for Bangladeshi households.”

RESPONSE: As the reviewer suggests we have incorporated this aspect in the discussion section. 

“Sixth, this study does not analyze medicines prices or price elasticity as a factor affecting medicines out-of-pocket expenditure because information on medicine prices is not collected as part of the survey. Linking outside data source with the data set used in this study is challenging as individuals in the survey do not report on specific products that they purchased.”

Moreover, it is noteworthy to mention to the reviewer that medicine price information collected through market studies often do not reflect the prices patients pay for their medicines.

Ajantha Sisira Kumara: “The covariates that the paper considered represent only the demand-side of medicines. The paper misses the story coming from supply-side factors in determining OOP expenses on medicines. Availability of public/private avenues for treatments, proximity to such avenues, health insurance coverage, facilities available in public/private avenues, influence of doctors are some of supply-side factors. Can you strengthen the analysis by incorporating the information pertaining to those supply-side factors? Accordingly, you may strengthen the sections of discussions and conclusions by describing the situation and the role of such supply-side factors in determining OOP expenses on medicines.”

RESPONSE: The reviewer is correct that this analysis focuses on the demand-side. It would be very interesting to link the demand side data with the supply side. However, this is outside the scope of this manuscript. In addition, it is noteworthy, that linking supply side factors such as availability of medicines in public and private sector and prescriber behavior in the catchment area of individual households are often absent or not updated. We have included this limitation in the discussion sector. 

“Finally, this study focuses on determinants of out-of-pocket expenditure that are recorded within this survey. We did not link other databases to study determinants of expenditure such as distance to the nearest government/private hospital, nearest public/private clinics, nearest public/private dispensary, and availability of doctors, specialists, dentists per 1,000 of population as they might have an explanatory power.”

4th Revision July 2022

Ajantha Sisira Kumara: “However, I still believe that you can calculate income elasticity of medical expenditure at the household-level. For this purpose, you no need to have the data pertaining to unit prices of medicine. I hope your dataset has household income or otherwise, it can be proxied by household expenditure.

Further, I still believe that you can incorporate health-related supply-side factors to calculate their impact on OOP expenditure on medicine. Those data are available at the country's macro-level and can be collected from annual reports of relevant ministries.

I firmly believe that the paper will be in a better shape after you address at least these two issues. Congratulations!”

---

## [Editor Report · Decision Letter 4]

2 Sep 2022

Out-of-pocket expenditure on medicines in Bangladesh: an analysis of the national household income and expenditure survey 2016-17

PONE-D-21-14254R4

Dear Dr. Wirtz,

We’re pleased to inform you that your manuscript has been judged scientifically suitable for publication and will be formally accepted for publication once it meets all outstanding technical requirements.

Kind regards,

Mohammad Bellal Hossain

Academic Editor

PLOS ONE
---

## [Editor Report · Acceptance letter]

6 Sep 2022

PONE-D-21-14254R4 

Out-of-pocket expenditure on medicines in Bangladesh: an analysis of the national household income and expenditure survey 2016-17 

Dear Dr. Wirtz:

I'm pleased to inform you that your manuscript has been deemed suitable for publication in PLOS ONE. Congratulations! Your manuscript is now with our production department. 

Kind regards, 

on behalf of

Dr. Mohammad Bellal Hossain 

Academic Editor

PLOS ONE